# Understanding the Prompt Sensitivity

## Abstract

Prompt sensitivity, which refers to how strongly the output of a large language model (LLM) depends on the exact wording of its input prompt, raises concerns among users about the LLM's stability and reliability. In this work, we consider LLMs as multivariate functions and perform a first-order Taylor expansion, thereby analyzing the relationship between prompts, their gradients, and the logit of the model's next token. Furthermore, according to the Cauchy–Schwarz inequality, the logit difference can be upper bounded by the product of the gradient norm and the norm of the difference between the prompts' embeddings or hidden states. Our analysis allows a general interpretation of why current transformer-based autoregressive LLMs are sensitive to prompts with the same meaning. In particular, we show that LLMs do not internally cluster similar inputs like smaller neural networks do, but instead disperse them. This dispersing behavior leads to an excessively large upper bound on the logit difference between the two prompts, making it difficult to be effectively reduced to zero. In our analysis, we also show which types of meaning-preserving prompt variants are more likely to introduce prompt sensitivity risks in LLMs. Our findings provide crucial evidence for interpreting the prompt sensitivity of LLMs. Code for experiments is available in the supplementary materials.

## 1 Introduction

Large language models (LLMs) usually show sensitivity to even minor variations in prompts, such as wording, prompt template, or even minor spelling errors, although these variations do not change the meaning of the prompt (Chatterjee et al., 2024). This phenomenon can be described as LLMs' prompt sensitivity, which can amplify the output variance, making the model's output unreliable. To quantify this effect, researchers (Zhuo et al., 2024; Chatterjee et al., 2024) have made considerable efforts to assess the sensitivity of LLMs to minor variations in prompts. Also, Sun et al. (2024) have attempted to improve the generalization ability of LLMs through reinforcement learning from human feedback (RLHF; Christiano et al., 2017) or instruction tuning (Wei et al., 2021). However, even minor changes such as prompt formatting to the wording of the prompts still can lead to the prompt sensitivity of these models (Sclar et al., 2024).

Although prompt sensitivity in LLMs is frequently highlighted, its generation mechanism remains poorly understood. For example, we still do not understand why a set of meaning-preserving prompts can yield completely different outputs by an LLM. This open issue leads to a lack of credibility in previous benchmark-based prompt sensitivity evaluations (Zhuo et al., 2024; Chatterjee et al., 2024) and the arbitrary practice of fine-tuning LLMs by increasing training samples (Liu et al., 2025; Dong et al., 2024). Previous studies (Zhuo et al., 2024; Chatterjee et al., 2024) calculate a metric to represent a model's sensitivity to wording changes in prompts based on its output. However, they make only limited contributions to understanding the prompt sensitivity of LLMs and fail to guide fundamental breakthroughs.

Contrary to previous studies, we aim to understand the prompt sensitivity of LLMs using a mathematical analysis method: Taylor expansion (Taylor, 1715). In this study, we focus on the current popular transformer-based LLMs. Specifically, we formalize the transformer blocks of LLMs as a continuous multivariate function, which outputs the logit of the model's next token. The hidden states are responsible for converting the prompts in discrete space into the continuous representation space. The hidden states of the prompt can be regarded as the multivariate input of the function. Then, we use the first-order Taylor expansion of this function to connect the hidden states of the

prompt with the output logit. Furthermore, formalizing LLMs as functions and performing Taylor expansion allows us to make connections between the gradients and hidden states of the model inputs. Monitoring their changes across the model's layers can help us understand the underlying mechanism of the prompt sensitivity of LLMs.

Moreover, we provide a novel perspective on explaining the prompt sensitivity of LLMs. Our analysis starts with an image classification task. We observe that ResNet (He et al., 2016) internally produces a clustering behavior to achieve high accuracy. However, we find that this behavior does not exist within transformer-based LLMs. By applying the Cauchy–Schwarz (Cauchy, 1821; Schwarz, 1890) inequality to transform the Taylor expansion, we reveal that this clustering behavior influences the upper bound of the logit difference between prompts. In other words, without the clustering behavior, the upper bound on the logit difference between prompts will hardly be sufficiently small to be effectively reduced to zero. In addition, we analyze different types of meaning-preserving prompt variants, such as modifying tokens in the first or latter half of the prompt, or modifying token order to create misalignment within the prompt. Our experiments indicate that modifying the first half of a prompt carries a higher risk of prompt sensitivity than modifying the latter half, with more token misalignments posing a greater risk than fewer token misalignments. Overall, token misalignments present a higher risk of prompt sensitivity than token modifications.

## 2 NEURAL NETWORKS ARE FUNCTIONS

A neural network is a mathematical relationship that maps inputs to outputs (LeCun et al., 2015; Nielsen, 2015; Goodfellow et al., 2016). If the input is a vector $x \in \mathbb{R}^d$ and the output is a scalar $y \in \mathbb{R}$, then a single-layer neural network can be represented as:

$$y = \sigma(w^\top x + b) \tag{1}$$

where $w$ is the weight vector, $b$ is the bias, and $\sigma(\cdot)$ is the activation function, such as sigmoid (Rumelhart et al., 1986), ReLU (Nair & Hinton, 2010; Glorot et al., 2011), etc. If we consider this neural network as a function $y = f(x)$. The one-time inference using this neural network can be interpreted as input vector $x$ to the function $f(x)$, outputting the scalar $y$. In this section, we start by explaining why deep neural networks are composite functions. Then, we introduce intra-class mean distances, a simple representation of space distances. Finally, we interpret why deep neural networks can perform classification tasks from an interesting perspective: that a neural network is a function.

**Deep neural networks are compositions of functions.**
A deep neural network defines a function as a composition of simpler functions. In particular, it is composed of layer-by-layer composites of affine transformations and activation functions (Cybenko, 1989; Hornik et al., 1989; Murphy, 2012). Formally, the affine transformation of the layer $l$ is:

$$A_l(x) = W_l x + b_l \tag{2}$$

Figure 1: ResNet example.

where $x \in \mathbb{R}^{d_{l-1}}$ is the output vector of layer $l-1$, $W_l \in \mathbb{R}^{d_l \times d_{l-1}}$ is the weight of the layer $l$, and $b$ is the bias of layer $l$. Then, the affine transformation $A_l(x)$ is composed using the activation function $\sigma_l$ as follows:

$$g_l = \sigma_l \circ A_l \tag{3}$$

The general mapping of the deep neural network of layer $L$ is as follows:

$$F = g_L \circ g_{L-1} \circ \cdots \circ g_1 \tag{4}$$

where the composite function $F$ is a continuous mapping from the input space to the output space (Hornik et al., 1989; Goodfellow et al., 2016).

Figure 1 shows the structure of a neural network used for image classification, with ResNet (He et al., 2016) connecting a projection layer and a fully connected layer. In the upper case, we consider "Stem" as the feature processing part $\mathcal{F}_p$, which outputs the feature map $\mathbf{c}_s$; the rest of the network serves as the function part $\mathcal{F}_f$, taking $\mathbf{c}_s$ as input and producing the result $\hat{y}$. Alternatively, in the

bottom case, we consider "Stage 4" and previous stages as the feature processing part $\mathcal{F}_p$, which outputs the feature map $\mathbf{c}_4$; the rest of the network is the function part $\mathcal{F}_f$, taking $\mathbf{c}_4$ as input and producing the result $\hat{y}$. In this way, we can split the composite function F into two parts: the data processing part $\mathcal{F}_p$ and the functional part $\mathcal{F}_f$. Thus, we can interpret the behavior of neural networks from a functional perspective.

**Intra-class compactness.** Intra-class compactness (Yan et al., 2020) refers to how close or tightly clustered the samples or data points of the same class are in the feature space. Typically, an ideal classifier requires ensuring high intra-class compactness. To remove the influence of vector dimension on distance metrics, we first perform $L^2$ normalization on the feature vectors, then use the Euclidean distance between the normalized vectors as the metric:

$$d(\boldsymbol{x}_i, \boldsymbol{x}_j) = \|\boldsymbol{x}_i - \boldsymbol{x}_j\| \tag{5}$$

As all vectors are normalized to the unit hypersphere, this distance reflects only directional differences and is equivalent to cosine similarity, e.g., $\|\boldsymbol{x}_i - \boldsymbol{x}_j\| = \sqrt{2 - 2\cos\theta_{ij}}$, where $\theta_{ij}$ is the angle between the two vectors. We denote the sample set for class $c$ as $\mathcal{J}_c$. We use the intra-class mean distance to measure intra-class compactness. The distance of samples in class $c$ is defined as follows:

$$D_{\text{intra}}^{(c)} = \frac{1}{|\mathcal{J}_c|(|\mathcal{J}_c| - 1)} \sum_{\boldsymbol{x}_i, \boldsymbol{x}_j \in \mathcal{J}_c} d(\boldsymbol{x}_i, \boldsymbol{x}_j) \tag{6}$$

We use the average of the distances over all classes as the metric for intra-class compactness:

$$D_{\text{intra}} = \frac{1}{|\mathcal{C}|} \sum_{c \in \mathcal{C}} D(c) \tag{7}$$

where $\mathcal{C}$ is the class set and $|\mathcal{C}|$ denotes the total number of classes. Enhancing intra-class compactness, i.e., decreasing $D_{\text{intra}}$, can improve the neural network's classification performance (Liu et al., 2016; Yan et al., 2020).

**Intra-class mean distance of ResNet on CIFAR-10.** To illustrate the internal behavior of neural networks while performing classification tasks. We investigate how intra-class compactness of the feature maps changes across each stage of a neural network. We take the application of ResNet (we pick ResNet-101; He et al., 2016) on the CIFAR-10 dataset (Krizhevsky et al., 2009) as an example. ResNet is typically divided into four stages (see Figure 1), each stage consisting of multiple residual blocks stacked together, and outputs the feature map $\mathbf{c}_{\{1,2,3,4\}}$ of the input image. Here, we focus on the stage level of ResNet to analyze the feature maps output at each stage, the input feature map $\mathbf{c}_s$ output by the stem.

We trained this neural network on the CIFAR-10 dataset for 100 epochs, achieving the highest F1 score on the testing set at epoch 91.[1] As shown in Figure 2, we compare the $D_{\text{intra}}$ of feature maps across three epochs: the early training epoch (epoch 1), the mid-training epoch (epoch 45), and the late training epoch (epoch 90). A low $D_{\text{intra}}$ indicates high intra-class compactness. In this example, we consider the ResNet part of the neural network as the feature processing part $\mathcal{F}_p$ and show the trend of intra-class mean distances of the feature maps at the $\mathcal{F}_p$ part. The projection and fully connected layers as the function part $\mathcal{F}_f$, and $\mathbf{c}_4$ are the output of the feature processing part $\mathcal{F}_p$ as the input of the function part $\mathcal{F}_f$. We find that in the feature processing part $\mathcal{F}_p$ exhibits a clustering behavior towards the feature maps. In particular, this clustering behavior becomes weaker on feature map $\mathbf{c}_4$ at earlier epochs (when F1 scores are relatively low). However, at later epoch,

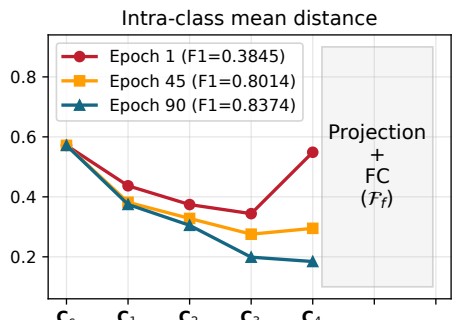

Figure 2: Intra-class mean distances for CIFAR-10 across different training epochs.

---

[1]For more hyperparameters, see Appendix B.1.

when F1 scores are relatively high, this clustering behavior persists on feature map $\mathbf{c}_4$. From the function perspective, clustering brings samples of the same class closer together, while continuous functions produce similar outputs for similar inputs. In other words, the classification accuracy of the functional part is influenced by the clustering effectiveness of the feature processing part. In this paper, based on the above analysis, we formulate LLMs as functions and investigate their prompt sensitivity using Taylor expansion.

## 3 INTERPRETATION OF PROMPT SENSITIVITY

The prompt sensitivity of LLMs usually refers to minor variations in prompts causing LLMs to respond with different results (Zhuo et al., 2024; Chatterjee et al., 2024). In this paper, we narrow it down to describe "how prompt $p_0$ and its meaning similar prompt $p_1$ cause the LLMs to respond with different logits of the model's next token $y_t$." The natural language prompts or their tokenized tokens reside in a discrete space, while their embeddings represented by the embedding layer or hidden states output by a specific transformer block can be regarded as variables in the continuous representation space.

### 3.1 LLMs ARE MULTIVARIABLE FUNCTIONS

In § 2, we observe that ResNet exhibits clustering behavior to achieve significantly higher accuracy and F1 score. We split the neural network into a feature processing part $\mathcal{F}_p$ and a function part $\mathcal{F}_f$. From this perspective, we interpret the neural network as follows: the $\mathcal{F}_p$ part clusters the input samples, while the $\mathcal{F}_f$ part classifies the clustered samples. We also point out that any layers in a neural network can be regarded as a function, with the previous layers serving as the feature processing part for the input of the function part. In this section, we generalize this interpretation to transformer-based LLMs. Unlike classification neural networks, which project the feature representations into a class space, LLMs project the feature representations into a vocabulary space to predict the next token.

In LLMs' inference stage, when an LLM predicts the next token, it first maps the input tokens into embeddings by the embedding layer and adds positional encodings to form a sequence representation. Then, the sequence representation passes through several transformer blocks sequentially. In the self-attention module of each transformer block, a causal mask is applied to block tokens to the right of the current position, ensuring that each current position only depends on the content to its left. In this way, each position ultimately obtains a hidden state vector that contains only the prefix information. When the model processes the entire input sequence, it predicts the next token using the hidden state of the last position. This hidden state is projected to the vocabulary space via the output layer (typically a linear layer and softmax).

Now, suppose we input a prompt containing $L$ tokens into an LLM. The model maps each token in this prompt to a $D$-dimensional embedding. Following the analysis in § 2, we can consider the model's embedding layer as the feature processing part $\mathcal{F}_p$, while the remaining transformer blocks and output layer as the part of a multivariate function $\mathcal{F}_f$ accepting an $L \times N$ input. The logit of the model's next token is the output of function $\mathcal{F}_f$. Alternatively, the embedding layer and any preceding blocks can be considered as the feature processing part $\mathcal{F}_p$, with the remaining blocks as the function part $\mathcal{F}_f$. In this case, the hidden states output by the feature processing part $\mathcal{F}_p$ is the input of the function part $\mathcal{F}_f$. Ideally, to achieve approximate output logits, LLMs should cluster the hidden states of meaning-preserving prompts at any layer.

### 3.2 TAYLOR EXPANSION OF LLMs

Suppose we split an LLM into a feature processing part $\mathcal{F}_p$ and a function part $\mathcal{F}_f$. The hidden states output by $\mathcal{F}_p$ are denoted as $\mathbf{z}$, which serve as input to $\mathcal{F}_f$. We select the output of the output layer during training, $\log \pi(y_t|\mathbf{z})$, as the output value of the function. Here, $\pi = \mathrm{softmax}(\mathbf{z}')$ and $\mathbf{z}'$ are the logits output by the LLM. Formally, we can express the relationship between the hidden states $\mathbf{z}_0$ and $\mathbf{z}_1$ of two meaning-preserving prompts by Taylor expansion[2] as follows:

---

[2] Appendix A provides the first-order Taylor expansion for both univariate and multivariate cases.

$$\underbrace{\log \pi(y_t|\mathbf{z}_1)}_{1\times 1} - \underbrace{\log \pi(y_t|\mathbf{z}_0)}_{1\times 1} = \underbrace{\nabla_{\mathbf{z}} \log \pi(y_t|\mathbf{z}_0)^\top}_{1\times[L\times D]} \underbrace{(\mathbf{z}_1 - \mathbf{z}_0)}_{[L\times D]\times 1} + \mathcal{O}(\|\mathbf{z}_1 - \mathbf{z}_0\|^2) \qquad (8)$$

where $1\times[L\times D]$ or $[L\times D]\times 1$ means that the gradient matrix $\nabla_{\mathbf{z}}\pi(y_t|\mathbf{z}_0)^\top \in \mathbb{R}^{L\times D}$ or difference matrix $\mathbf{z}_1 - \mathbf{z}_0 \in \mathbb{R}^{D\times L}$ can be considered as a flattened matrix with shapes $1\times[L\times D]$ or $[L\times D]\times 1$ and $L$ is the prompt length and $D$ is the dimension of the model's hidden layer. $\mathcal{O}(\|\mathbf{z}_1 - \mathbf{z}_0\|^2)$ is the remainder term of the Taylor expansion.

The difference matrix $\Delta\mathbf{z} = \mathbf{z}_1 - \mathbf{z}_0$ represents the difference between the feature representations of the two prompts. It is calculated through element-wise subtraction, thus capturing not only semantic differences between the two prompts but also variations in their expressive styles. In real-world scenarios, as the number of tokens in two prompts may be different, the shapes of $\mathbf{z}_0$ and $\mathbf{z}_1$ may also be different. If $\mathbf{z}_0$ and $\mathbf{z}_1$ have different shapes, we will use $\mathbf{z}_0$'s shape as the standard to pad $\mathbf{z}_1$ with zero vectors or trim it accordingly. This hard alignment method might more or less overestimate the model's prompt sensitivity. We will analyze this impact in § 4.

### 3.3 Upper Bound

From the properties of Taylor expansion, we know that when the distance of $\mathbf{z}_0$ and $\mathbf{z}_1$ is sufficiently close, the remainder term $\mathcal{O}(\|\mathbf{z}_1 - \mathbf{z}_0\|^2)$ will vanish faster than $\|\mathbf{z}_1 - \mathbf{z}_0\|^2$ as $\mathbf{z}_1 \to \mathbf{z}_0$. Based on this condition, we rewrite Eq. (8) in the following form:

$$\log \pi(y_t|\mathbf{z}_1) - \log \pi(y_t|\mathbf{z}_0) \approx \nabla_{\mathbf{z}} \log \pi(y_t|\mathbf{z}_0)^\top (\mathbf{z}_1 - \mathbf{z}_0) \qquad (9)$$

Then, we obtain the following inequality by calculating the L2 norm:

$$|\log \pi(y_t|\mathbf{z}_1) - \log \pi(y_t|\mathbf{z}_0)| \leq \|\nabla_{\mathbf{z}} \log \pi(y_t|\mathbf{z}_0)\| \cdot \|\mathbf{z}_1 - \mathbf{z}_0\| \qquad (10)$$

This inequality tells us that $|\Delta \log \pi(y_t|\mathbf{z})| = |\log \pi(y_t|\mathbf{z}_1) - \log \pi(y_t|\mathbf{z}_0)|$ has an upper bound $\|\nabla_{\mathbf{z}} \log \pi(y_t|\mathbf{z}_0)\| \cdot \|\mathbf{z}_1 - \mathbf{z}_0\|$. Here, $\|\cdot\|$ actually is the Frobenius norm. As in Eq. (8) we convert the matrix into a one-dimensional vector, it is written here as the L2 norm. If $\|\nabla_{\mathbf{z}} \log \pi(y_t|\mathbf{z}_0)\| \cdot \|\mathbf{z}_1 - \mathbf{z}_0\|$ is significantly small, $|\log \pi(y_t|\mathbf{z}_1) - \log \pi(y_t|\mathbf{z}_0)|$ can be approximated as 0, meaning the two meaning-preserving prompts receive equal logits of the model's next token.

**Calculate the gradient.** We represent the gradient matrix as follows:

$$\nabla_{\mathbf{z}} \log \pi(y_t|\mathbf{z})^\top = [G(\mathbf{z}[1,:]), \dots, G(\mathbf{z}_0[L,:])] \qquad (11)$$

Each row $G(\mathbf{z}[i,:]) \in \mathbb{R}^D$ of the gradient matrix represents the gradient vector of each token of the prompt. The gradient for the $j$-th dimension of the $i$-th token is calculated as follows:

$$g(\mathbf{z}[i,j]) = \nabla_{\mathbf{z}[i,j]} \log \pi(y_t|\mathbf{z}) \qquad (12)$$

The gradient $g(\mathbf{z}[i,j])$ is the gradient of the logits of the model's next token, usually named saliency score (Simonyan et al., 2013; Li et al., 2016). Unlike Yin & Neubig (2022), who use the L1 norm to calculate the saliency score for each input token, we take the Frobenius norm of the gradient matrix to obtain the saliency score of input $\mathbf{z}$ as follows:

$$S_{GN}(\mathbf{z}) = \|\nabla_{\mathbf{z}} \log \pi(y_t|\mathbf{z})\|_F = \sqrt{\sum_{i,j} |g(\mathbf{z}[i,j])|^2} \qquad (13)$$

$S_{GN}(\mathbf{z})$ is the overall contribution of $\mathbf{z}$ to the logit of the model's next token.

### 4 Experimental Verifications

In this section, we verify our analytical results in practical settings. We consider four datasets commonly used to evaluate prompt sensitivity (Zhuo et al., 2024; Chatterjee et al., 2024), ARC Challenge (Clark et al., 2018), CommonSenseQA (Talmor et al., 2019), MMLU (Hendrycks et al., 2021), and OpenBookQA (Mihaylov et al., 2018). We randomly select 500 examples to create

our test set from each dataset. Each sample in our test set is a multiple-choice question with a correct answer. We consider 12 prompt templates[3] provided in Zhuo et al. (2024)'s work to create the meaning-preserving prompts for LLMs. We perform all our experiments on two model series: `Qwen1.5-0.5B/1.8B/4B` (Bai et al., 2023) and `Llama3.2-1B/3B` (Touvron et al., 2023). Unless otherwise specified, we only report experimental results for `Llama3.2-3B` on the ARC Challenge dataset in this section. For detailed experimental results and further discussion regarding other models and datasets, please refer to Appendix B.

## 4.1 PERTURBATION ANALYSIS

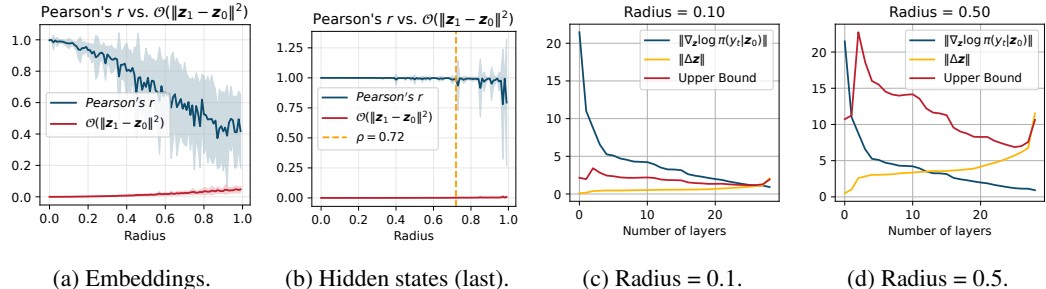

(a) Embeddings.    (b) Hidden states (last).    (c) Radius = 0.1.    (d) Radius = 0.5.

Figure 3: Key results of perturbation verification for `Llama3.2-3B` on ARC Challenge dataset: (a) and (b) are the Pearson's $r$ between $\Delta \log \pi(y_t|\mathbf{z})$ and $\nabla_\mathbf{z} \log \pi(y_t|\mathbf{z}_0)^\top \Delta \mathbf{z}$ of embeddings and the last layer's hidden states respectively. (c) and (d) are the variations of $\|\nabla_\mathbf{z} \log \pi(y_t|\mathbf{z}_0)\|$, $\|\Delta \mathbf{z}\|$, and their product across model's layers when the perturbation radius is 0.1 and 0.5 respectively (full results in Appendix B.4 and B.5).

Eq. (8) describes how the difference between the output logits of two hidden states $\mathbf{z}_0$ and $\mathbf{z}_1$ can be approximated by the difference between the gradient matrix dot products of $\Delta \mathbf{z} = \mathbf{z}_1 - \mathbf{z}_0$. The Taylor expansion converges in some open neighborhood of $\mathbf{z}_0$, which can be written as:

$$B_r(\mathbf{z}_0) = \{\mathbf{z}_1 \in \mathbb{R}^D \mid \|\mathbf{z}_1 - \mathbf{z}_0\| < \rho\} \tag{14}$$

where $\rho$ is the perturbation radius. We start by analyzing the relationship between the term $\Delta \log \pi(y_t|\mathbf{z})$ and $\nabla_\mathbf{z} \log \pi(y_t|\mathbf{z}_0)^\top \Delta \mathbf{z}$ to verify the validity of Eq. (8). First, we consider the LLM's embedding layer as the feature processing part $\mathcal{F}_p$ and all the transformer blocks as the function part $\mathcal{F}_f$. We randomly pick a prompt as the seed prompt $p_s$, the embedding vectors of the prompt $p_s$ are $\mathbf{z}_0$. Then, we randomly perturb the embedding vectors $\mathbf{z}_0$ to obtain the perturbed embedding vectors $\mathbf{z}_1$. We create 100 randomly perturbed embedding vectors as the input to the function by constraining the norm between $\mathbf{z}_0$ and $\mathbf{z}_1$ to be less than the perturbation radius $\rho$. The range of our perturbation radius $\rho$ is from 0 to 1, with a step size of 0.01. We calculate $\Delta \log \pi(y_t|\mathbf{z})$ and $\nabla_\mathbf{z} \log \pi(y_t|\mathbf{z}_0)^\top \Delta \mathbf{z}$ for these 100 embedding vectors with $\mathbf{z}_0$. We repeated the experiment on 10 seed prompts and calculated the average results.

As shown in Figure 3a, we report the Pearson's $r$ (Pearson, 1895) between $\Delta \log \pi(y_t|\mathbf{z})$ and $\nabla_\mathbf{z} \log \pi(y_t|\mathbf{z}_0)^\top \Delta \mathbf{z}$ for the 100 perturbed samples under different perturbation radius $\rho$, using `Llama3.2-3B`. A high Pearson's $r$ indicates that the first-order Taylor expansion provides a good linear approximation to the function represented by the LLM. We observe that on the embedding vectors, the Pearson's $r$ gradually decreases as the perturbation radius increases, indicating that the first-order linear approximation progressively fails and the influence of higher-order terms becomes increasingly evident.

However, as shown in Figure 3b, the correlation between $\Delta \log \pi(y_t|\mathbf{z})$ and $\nabla_\mathbf{z} \log \pi(y_t|\mathbf{z}_0)^\top \Delta \mathbf{z}$ in the hidden states output by the last transformer block remains at a high and stable level when the perturbation radius is less than 0.72. This indicates that within a small perturbation range, the linear approximation can effectively explain the variation in $\Delta \log \pi(y_t|\mathbf{z})$. As the radius continues to increase, the correlation begins to fluctuate. This phenomenon might stem from two reasons: *(1)* The mapping between the last hidden states and the output is relatively closer to linear (typically

---

[3]All templates are provided in Appendix B.2.

a linear layer followed by softmax). Therefore, within a small perturbation range, the first-order approximation holds well, manifesting as a high and stable Pearson's $r$. As the perturbation radius increases further, nonlinear effects such as softmax begin to become significant, causing the correlation to fluctuate. *(2)* From the perspective that LLMs are functions, after processing by the feature processing part $\mathcal{F}_p$, perturbed samples have been "regularized" or "clustered" in the latent space, allowing changes in $\Delta \log \pi(y_t|\mathbf{z})$ to be largely explained by gradient inner products.

The first reason is obvious, as LLMs are stacks of multiple non-linear transformer blocks. However, regarding the second reason, do LLMs actually perform internal clustering of similar meaning inputs? In Figure 3c and 3d, we compare the variations of $\|\nabla_{\mathbf{z}} \log \pi(y_t|\mathbf{z}_0)\|$, $\|\Delta \mathbf{z}\|$, and their product (i.e., the upper bound of $\Delta \log \pi(y_t|\mathbf{z})$) across the model's layers. We find that although the gradient gradually decreases with each layer, the distance between perturbation samples and the seed sample in the representation space continues to increase, rising sharply at the last hidden states. The result is that the upper bound of $\Delta \log \pi(y_t|\mathbf{z})$ (the red line) fails to converge to sufficiently low values, making it difficult for $\Delta \log \pi(y_t|\mathbf{z})$ to approach zero. This indicates that the internal representations of LLMs do not exhibit clustering behavior similar to traditional neural networks. Previous studies have revealed clustering behaviors in transformer-based LLMs (Phang et al., 2021; Wu & Varshney, 2025), but these behaviors are typically limited to clustering the same task samples. In contrast, our analysis requires LLMs to cluster meaning-preserving prompts together. In light of the findings of Wu & Varshney (2025), we argue that current mainstream LLMs exhibit insufficient clustering behavior to entail $\|\nabla_{\mathbf{z}} \log \pi(y_t|\mathbf{z}_0)\| \cdot \|\Delta \mathbf{z}\|$ as an effective upper bound.

In summary, we have the following interpretation for prompt sensitivity of LLMs. First, LLMs do not exhibit the clustering behavior that is found in traditional neural networks. However, this clustering behavior serves as crucial evidence that neural networks can accurately perform classification tasks. Secondly, as LLMs tend to pull meaning-preserving prompts farther apart in the representation space, this leads to giving $\Delta \log \pi(y_t|\mathbf{z})$ a large upper bound $\|\nabla_{\mathbf{z}} \log \pi(y_t|\mathbf{z}_0)\| \cdot \|\Delta \mathbf{z}\|$. This makes it impossible to approximate $\Delta \log \pi(y_t|\mathbf{z})$ to zero via the upper bound. In other words, because LLMs do not exhibit clustering behavior for meaning-preserving prompts, they can only learn each sample individually during training. This cannot guarantee that the model fits meaning-preserving prompts to the same degree, leading to different outputs during inference.

## 4.2 REAL-WORLD DATASET VALIDATION

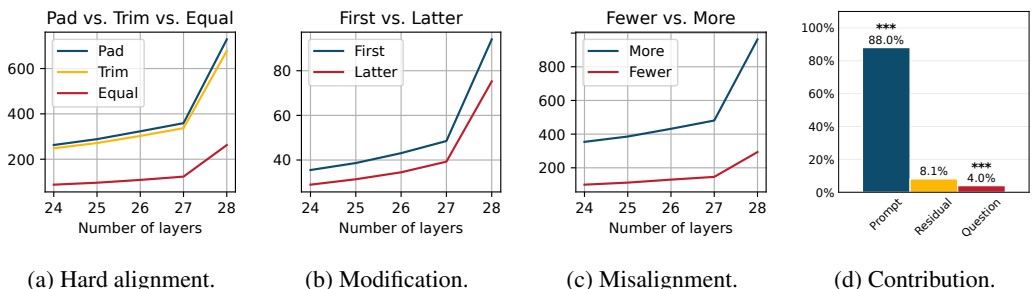

(a) Hard alignment.    (b) Modification.    (c) Misalignment.    (d) Contribution.

Figure 4: Key results of `Llama3.2-3B` on ARC Challenge dataset: (a), (b), and (c) are the $\|\Delta \mathbf{z}\|$ of the last 5 layers of `Llama3.2-3B` under different conditions. (d) is the comparison of contributions from the prompt template and the question. The asterisks (***) indicate $p$-value$<0.001$ (full results in Appendix B.6).

In § 3.2, we mention that when calculating Eq. (8), if $\mathbf{z}_0$ and $\mathbf{z}_1$ have different shapes, we will use $\mathbf{z}_0$'s shape as the standard to pad $\mathbf{z}_1$ with zero vectors or trim it accordingly. In this section, we first discuss the impact of this hard alignment method. In particular, we use our randomly selected test set. We fill each sample into 12 prompt templates to obtain 12 meaning-preserving prompts. For each sample's 12 meaning-preserving prompts, we pair them in all possible combinations and calculate the average $\|\Delta \mathbf{z}\|$ under three conditions: *pad*, *trim*, and *equal*.[4] Figure 4a shows the

---

[4]We refer to the number of tokens of the prompt $\mathbf{z}$ as $\text{len}(\mathbf{z})$. Therefore, *pad* means $\text{len}(\mathbf{z}_0) > \text{len}(\mathbf{z}_1)$, *trim* means $\text{len}(\mathbf{z}_0) < \text{len}(\mathbf{z}_1)$, and *equal* means $\text{len}(\mathbf{z}_0) = \text{len}(\mathbf{z}_1)$.

average results of `Llama3.2-3B` on the ARC Challenge dataset's 500 samples. We find that the trend of $\|\Delta z\|$ is the same as the perturbation experiment results in § 4.1, regardless of whether *pad*, *trim*, or *equal* conditions. However, *pad* and *trim* yield relatively higher $\|\Delta z\|$ than *equal*. We also note that even the relatively low $\|\Delta z\|$ for *equal* is significantly higher than the $\|\Delta z\|$ for the perturbation radius of 0.1 and 0.5 in Figure 3c. This indicates that prompt templates designed for real-world scenarios are highly diverse, making it challenging for LLMs to generate consistent outputs based on these templates.

To investigate which specific types of prompts may lead to higher prompt sensitivity, we create four types of prompts for quantitative analysis. All prompt templates are modified from a seed prompt template. Following Zhuo et al. (2024)'s setting, we create three prompt templates for each prompt type. Our four types of prompt templates are as follows:[5]

1. Modification first: replace one token in the first half of the seed prompt template with a synonymous token.

2. Modification latter: replace one token in the latter half of the seed prompt template with a synonymous token.

3. Misalignment fewer: modify a few tokens in the seed prompt template to make them slightly token misalignment.

4. Misalignment more: modify the token order in the seed prompt template to make them significant token misalignment.

We randomly select 500 samples from each dataset for testing. For clarity, we refer to prompts generated using the seed template as $\mathbf{z}_{seed}$, and those created using our prompt templates 1 to 4 as $\mathbf{z}_{first}$, $\mathbf{z}_{latter}$, $\mathbf{z}_{fewer}$, and $\mathbf{z}_{more}$ respectively. From Eq. (10), we can know that when $\mathbf{z}_0$ remains constant, the upper bound of $\Delta \log \pi(y_t|\mathbf{z})$ is determined by $\|\Delta \mathbf{z}\|$. This implies that a higher $\|\Delta \mathbf{z}\|$ imposes less constraint on $\Delta \log \pi(y_t|\mathbf{z})$. Therefore, we calculate $\|\Delta \mathbf{z}\|$ between $\mathbf{z}_{seed}$ and our four types of prompts for comparison. Figure 4b shows the results on the last 5 layers of `Llama3.2-3B` on the ARC Challenge dataset. We find that $\mathbf{z}_{latter}$ achieves a lower $\|\Delta \mathbf{z}\|$ than $\mathbf{z}_{first}$, indicating that modifying tokens at the beginning of the prompt has a greater impact on the output of LLMs than modifying tokens at the end. This finding aligns with the recent work by Wu et al. (2025), which shows that causal masking inherently biases attention toward earlier positions, as tokens in deeper layers attend to increasingly more contextualized representations of earlier tokens. In addition, the results in Figure 4c indicate that despite preserving the meaning, changing the token order of prompts might lead to a greater impact.

The work by Wu & Varshney (2025) indicates that LLMs tend to cluster the same task samples. This inspires us to evaluate whether the outputs of LLMs are more influenced by the prompt template or the question itself. In particular, we construct an ordinary least squares regression model that uses different prompt templates and questions to predict the logit of the model's next token. Subsequently, we perform an analysis of variance (ANOVA) on the regression results to calculate each factor's contribution to the total variance and its statistical significance, and further determine each factor's proportion of contribution relative to the total sum of squares. Figure 4d shows the results of `Llama3.2-3B` on the ARC Challenge dataset. The prompt template is the primary factor explaining the logit variation, with an explanatory rate of 88.0%. Although the question factor explains only 4.0% of the logit variation, its effect is statistically significant, indicating that different questions still exert a systematic influence on logits. Meanwhile, the residual accounts for 8.1%, suggesting that a portion of the variation remains unexplained by either the prompt or question. This residual variance may arise from data noise or factors not modeled by the LLM.

## 5 RELATED WORK

### 5.1 PROMPT SENSITIVITY OF LLMS

LLMs have strong in-context learning capabilities (Brown et al., 2020), enabling them to perform diverse tasks based on prompts, often without requiring additional fine-tuning (Radford et al., 2019;

---

[5]For details of the prompt templates, please refer to Appendix B.3.

Raffel et al., 2020; Gao et al., 2021). However, the stability and reliability of this learning approach remain controversial (Weber et al., 2023). Existing studies indicate that model outputs are highly dependent on multiple factors, such as the choice and order of examples (Liu et al., 2022; SU et al., 2023; Lu et al., 2022; Zhao et al., 2021), the definition of input labels (Min et al., 2022), and the phrasing of prompts (Gu et al., 2023; Sun et al., 2024). Beyond these factors, LLMs exhibit extreme sensitivity to minor changes in prompt structure or phrasing, even when such alterations preserve semantic meaning. This phenomenon has been systematically explored in numerous studies (Voronov et al., 2024; Mizrahi et al., 2024), indicating that subtle modifications to prompts can significantly impact model outputs. Furthermore, to characterize and compare the prompt sensitivity of different models, numerous studies (Zhu et al., 2023; Zhuo et al., 2024; Chatterjee et al., 2024) have constructed specialized benchmarks to quantify and evaluate models' robustness to prompt perturbations. Contrary to previous work, this study attempts to represent LLMs as functions, leveraging Taylor expansion to explain the mechanism behind prompt sensitivity from the function perspective. It provides both theoretical foundations and empirical evidence to explain why LLMs exhibit prompt sensitivity.

## 5.2 LLMs as Functions

In recent years, some studies have attempted to characterize LLMs from the perspective of function mapping (Brown et al., 2020; Wei et al., 2022). This perspective abstracts LLMs as conditional probability distribution functions mapping input spaces to output spaces. In other words, given a prompt $x$, the model defines a distribution $P(y|x)$ for an output $y$. This functional representation facilitates a unified understanding of model behavior across different tasks and provides a theoretical framework for analyzing LLM generalization and robustness. Notably, it has also been shown that transformers themselves serve as universal approximators of sequence-to-sequence functions (Yun et al., 2020), further reinforcing the perspective that LLMs are functions. Building on this idea of function mapping, some studies consider prompt engineering as a design problem for function call interfaces, investigating how different prompt formats alter the properties of the function mapping (Liu et al., 2023). In our study, we consider LLMs as composite functions that can be split into the feature processing part and the function part between any transformer blocks. This split allows us to perform a Taylor expansion on any part of the models for analysis.

## 6 Conclusion and Limitations

Prompt sensitivity, which describes how LLMs produce different outputs in response to meaning-preserving prompts, raises user concerns about the stability and reliability of LLMs. To investigate the underlying mechanisms of prompt sensitivity and to better understand LLMs, we start by considering LLMs as multivariate continuous functions. We point out that improving classification accuracy requires the internal clustering behavior within neural networks. Then, we apply the first-order Taylor expansion to LLMs. By observing changes in hidden states across all layers, we find that transformer-based LLMs lack this clustering behavior, which leads to the models failing to approximate the difference of logits between meaning-preserving prompts to zero. We also note that modifying the first half of the prompt is more likely to trigger prompt sensitivity than modifying the latter half, and the risk increases with the number of misaligned tokens. Overall, misalignment of tokens poses a greater risk of prompt sensitivity than token modification. We also find that the prompt template has a greater impact on model output than the question.

One limitation of this work is the hard alignment method in Eq. (8), as mentioned in the paper, which might more or less overestimate the model's prompt sensitivity. Another limitation is that we only considered the logit of a single dimension for the model's next token, implicitly requiring the entire logit distribution of the next token to remain consistent across meaning-preserving prompts. This requirement poses a significant challenge for LLMs. Finally, we employed only a first-order Taylor expansion. Given that LLMs are naturally highly complex functions, this linear approximation may introduce some errors. As illustrated by the perturbation analysis in § 4.1, the Pearson's $r$, which reflects the accuracy of the linear approximation, gradually decreases as the perturbation radius increases. In the future, introducing higher-order Taylor expansions could be considered to achieve more precise approximations.

ETHICS STATEMENT

Our study aims to reveal the underlying mechanisms of prompt sensitivity of LLMs, thereby inspiring future works to develop more stable and reliable LLMs based on our findings. We acknowledge the contributions of previous efforts and cite relevant works in our paper.

REPRODUCIBILITY

To ensure the reproducibility of our findings, details of the experimental procedures are provided in Appendix B. Additionally, we will open-source our code on GitHub. These measures are intended to facilitate verification and replication of our results by other researchers in the field.

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

## A  TAYLOR EXPANSION

### A.1  BACKGROUND

The roots of Taylor expansion can be traced back to early thoughts on infinity, such as the paradoxes of divisibility proposed by the ancient Greek philosopher Zeno (Lindberg, 1992), as well as the "method of exhaustion" developed by Archimedes (Heath, 1981) and later by Liu Hui (Martzloff, 2007), which laid the foundation for approximating infinite processes through finite steps. In the 14th century, Indian mathematician Madhava of Sangamagrama and his successors in the Kerala school developed series expansions for functions such as sine, cosine, and arctangent, marking the earliest concrete examples of power series methods analogous to later Taylor expansions (Lindberg, 1992). In the 17th century, Newton and Gregory independently developed general methods for expanding functions (Inglis, 1940). Later, Brook Taylor first systematically proposed an expansion method applicable to general functions in 1715, forming the basis of today's Taylor expansions (Taylor, 1715). In our study, we consider LLMs as functions and employ first-order Taylor expansions to connect prompts, their gradients, and the logit of the model's next token, thereby analyzing the constraint relationships among them.

### A.2  THE FIRST-ORDER TAYLOR EXPANSION

In mathematics, the Taylor series or Taylor expansion of a function is an infinite sum of terms that are expressed in terms of the function's derivatives at a single point. The partial sum formed by the first $n+1$ terms of a Taylor series is a polynomial of degree n that is called the nth Taylor polynomial

of the function. Taylor polynomials are approximations of a function, which become generally more accurate as $n$ increases. The first-order Taylor expansion in one variable of $f(x)$ about $x = a$ is as follows:

$$f(x) = f(a) + f'(a)(x - a) + \mathcal{O}((x - a)^2) \quad (x \to a). \tag{15}$$

where $\mathcal{O}(x - a)$ indicates the infinitesimal term of higher order than $(x - a)$, and $x \to a$ indicates that this equality holds as $x$ approaches $a$. In other words, this expansion is a local approximation describing the behavior of $f(x)$ near $x = a$.

For more complex multivariate scenarios, we suppose $f : \mathbb{R}^n \to \mathbb{R}$ is differentiable at the point $\boldsymbol{a} = (a_1, a_2, \ldots, a_n)$. Then the first-order Taylor expansion of f at $\boldsymbol{x} = (x_1, x_2, \ldots, x_n)$ is:

$$f(\boldsymbol{x}) = f(\boldsymbol{a}) + \nabla f(\boldsymbol{a}) \cdot (\boldsymbol{x} - \boldsymbol{a}) + \mathcal{O}(\|\boldsymbol{x} - \boldsymbol{a}\|^2) \quad (\boldsymbol{x} \to \boldsymbol{a}). \tag{16}$$

where $\nabla f(\boldsymbol{a}) = \left( \frac{\partial f}{\partial x_1}(\boldsymbol{a}), \frac{\partial f}{\partial x_2}(\boldsymbol{a}), \ldots, \frac{\partial f}{\partial x_n}(\boldsymbol{a}) \right)$ is the gradient. In the expression $\mathcal{O}(\|\boldsymbol{x} - \boldsymbol{a}\|)$, the norm $\| \cdot \|$ can be any norm (such as the Euclidean norm (2-norm) or vector norm) on $\mathbb{R}^n$, because all norms in finite-dimensional spaces are equivalent. The $\mathcal{O}(\|\boldsymbol{x} - \boldsymbol{a}\|^2)$ means the remainder term that vanishes faster than $\|\boldsymbol{x} - \boldsymbol{a}\|^2$ as $\boldsymbol{x} \to \boldsymbol{a}$. The operator $\cdot$ denotes the dot product.

## B  MORE ABOUT EXPERIMENTS

### B.1  THE HYPERPARAMETERS FOR TRAINING RESNET.

To ensure stable optimization and efficient convergence of the `ResNet-101` network on the CIFAR-10 dataset, a carefully designed hyperparameter configuration scheme was employed during training.

As shown in Figure 1, our network architecture is a ResNet connected to a projection layer and a fully connected layer. This section provides more details. We preprocess images using the following pipeline before feeding them into ResNet:

```
transform = transforms.Compose([
    transforms.Resize(112),
    transforms.CenterCrop(112),
    transforms.ToTensor()
])
```

We project the 2048-dimensional features from ResNet's stage 4 output onto a 128-dimensional embedding space, then classify them using a fully connected (classification) layer. The specific network architecture is as follows:

```
proj = nn.Sequential(
    nn.Linear(2048, 512),
    nn.BatchNorm1d(512),
    nn.PReLU(),
    nn.Dropout(p=0.2),
    nn.Linear(512, 128)
)

clf = nn.Linear(128, num_classes)
```

Our experiment employs the cross-entropy loss function with the AdamW optimizer (Loshchilov & Hutter, 2017), using the macro F1 score as the primary evaluation metric. The training process utilizes a batch size of 128 and runs for 100 epochs.

### B.2  MEANING-PRESERVING PROMPT TEMPLATES

In this section, we provide the 12 prompt templates provided by Zhuo et al. (2024) mentioned in § 4. For multiple-choice questions with 4 options, the templates are shown in Table 1. We choose 12 prompts for experimentation to ensure data diversity and avoid inaccurate results caused by individual edge cases.

Table 1: The meaning-preserving prompt templates for ARC Challenge, MMLU, and OpenBookQA datasets. For the CommonSenseQA dataset, the number of options changes from four to five, so option 'E' should be added accordingly. Gray text indicates template slots that need to be replaced.

| | |
|---|---|
| prompt 1 | `{question}\nA. {A}\nB. {B}\nC. {C}\nD. {D}\nAnswer:` |
| prompt 2 | `Question:\n{question}\nA. {A}\nB. {B}\nC. {C}\nD. {D}\nAnswer:` |
| prompt 3 | `Question:\n{question} A. {A} B. {B} C. {C} D. {D}\nAnswer:` |
| prompt 4 | `Could you provide a response to the following question: {question} A. {A} B. {B} C. {C} D. {D}` |
| prompt 5 | `Please answer the following question:\n{question}\nA. {A}\nB. {B}\nC. {C}\nD. {D}` |
| prompt 6 | `Please address the following question:\n{question}\nA. {A}\nB. {B}\nC. {C}\nD. {D}\nAnswer:` |
| prompt 7 | `You are a very helpful AI assistant. Please answer the following questions: {question} A. {A} B. {B} C. {C} D. {D}` |
| prompt 8 | `As an exceptionally resourceful AI assistant, I'm at your service. Address the questions below:\n{question}\nA. {A}\nB. {B}\nC. {C}\nD. {D}` |
| prompt 9 | `As a helpful Artificial Intelligence Assistant, please answer the following questions\n{question} A. {A}\nB. {B}\nC. {C}\nD. {D}` |
| prompt 10 | `Could you provide a response to the following question: {question} A. {A} B. {B} C. {C} D. {D}\nAnswer the question by replying A, B, C or D.` |
| prompt 11 | `Please answer the following question:\n{question}\nA. {A}\nB. {B}\nC. {C}\nD. {D}\nAnswer the question by replying A, B, C or D.` |
| prompt 12 | `Please address the following question:\n{question}\nA. {A}\nB. {B}\nC. {C}\nD. {D}\nAnswer this question by replying A, B, C or D.` |

## B.3 MODIFICATION AND MISALIGNMENT PROMPT TEMPLATES

To evaluate which types of prompts may lead to higher prompt sensitivity, we create four prompt templates for quantitative analysis. These four prompt templates are shown in Table 2. These prompt templates are modified from a seed prompt template, which is: "`You are a very helpful AI assistant. Please answer the following questions:\nQuestion: {question}\nA. {A} B. {B} C. {C} D. {D}\nPlease choose the best`

```
option and respond only with the option of the correct answer (A,
B, C, or D).\nAnswer:"
```

Our experimental implementation process is as follows: We first randomly select 500 samples from each of the four datasets. We then combine these samples with both the seed prompt template and our modified 12 prompt templates, creating 6,500 prompts for each dataset. These prompts feed into the LLMs for testing.

Table 2: Our prompt templates for ARC Challenge, MMLU, and OpenBookQA datasets. For the CommonSenseQA dataset, the number of options changes from four to five, so option 'E' should be added accordingly. Gray text indicates template slots that need to be replaced. Green indicates the modified token in the first half of the prompt. Red indicates the modified token in the latter half of the prompt. Orange indicates the token causing the misalignment in the prompt. Blue indicates that the prompt is completely misaligned.

| Seed prompt | You are a very helpful AI assistant. Please answer the following questions:\nQuestion: {question}\nA. {A} B. {B} C. {C} D. {D}\nPlease choose the best option and respond only with the option of the correct answer (A, B, C, or D).\nAnswer: |
|---|---|
| Modification first | You are a very useful AI assistant. Please answer the following questions:\nQuestion: {question}\nA. {A} B. {B} C. {C} D. {D}\nPlease choose the best option and respond only with the option of the correct answer (A, B, C, or D).\nAnswer:

You are a very smart AI assistant. Please answer the following questions:\nQuestion: {question}\nA. {A} B. {B} C. {C} D. {D}\nPlease choose the best option and respond only with the option of the correct answer (A, B, C, or D).\nAnswer:

You are a very friendly AI assistant. Please answer the following questions:\nQuestion: {question}\nA. {A} B. {B} C. {C} D. {D}\nPlease choose the best option and respond only with the option of the correct answer (A, B, C, or D).\nAnswer: |
| Modification latter | You are a very helpful AI assistant. Please answer the following questions:\nQuestion: {question}\nA. {A} B. {B} C. {C} D. {D}\nPlease choose the best option and respond only with the option of the suitable answer (A, B, C, or D).\nAnswer:

You are a very helpful AI assistant. Please answer the following questions:\nQuestion: {question}\nA. {A} B. {B} C. {C} D. {D}\nPlease choose the best option and respond only with the letter of the correct answer (A, B, C, or D).\nAnswer:

You are a very helpful AI assistant. Please answer the following questions:\nQuestion: {question}\nA. {A} B. {B} C. {C} D. {D}\nPlease choose the best option and respond only with the choice of the correct answer (A, B, C, or D).\nAnswer: |
| Misalignment fewer | You are a very helpful AI assistant. Please answer the following questions:\nQuestion: {question}\nA. {A} B. {B} C. {C} D. {D}\nPlease choose the best option and respond only with the option of the answer (A, B, C, or D) below.\nAnswer:

You are a very helpful AI assistant. Please answer the following questions:\nQuestion: {question}\nA. {A} B. {B} C. {C} D. {D}\nPlease choose the best option and respond only with the option of the answer (A, B, C, or D) carefully.\nAnswer:

You are a very helpful AI assistant. Please answer the following questions:\nQuestion: {question}\nA. {A} B. {B} C. {C} D. {D}\nPlease choose the best option and respond only with the option of the answer (A, B, C, or D) now.\nAnswer: |
| Misalignment more | Please choose the best option and respond only with the option of the answer (A, B, C, or D) below.\nYou are a very helpful AI assistant. Please answer the following questions:\nQuestion: {question}\nA. {A} B. {B} C. {C} D. {D}\nAnswer:

Please choose the best option and respond only with the option of the answer (A, B, C, or D) carefully.\nYou are a very helpful AI assistant. Please answer the following questions:\nQuestion: {question}\nA. {A} B. {B} C. {C} D. {D}\nAnswer:

Please choose the best option and respond only with the option of the answer (A, B, C, or D) now.\nYou are a very helpful AI assistant. Please answer the following questions:\nQuestion: {question}\nA. {A} B. {B} C. {C} D. {D}\nAnswer: |

### B.4 More Results of Perturbation Analysis

Figures 5, 6, 7, and 7 illustrate the Pearson's $r$ between $\Delta \log \pi(y_t|\mathbf{z})$ and $\nabla_\mathbf{z} \log \pi(y_t|\mathbf{z}_0)^\top \Delta\mathbf{z}$ on Qwen1.5-0.5B, Qwen1.5-1.8B, Qwen1.5-4B, Llama3.2-1B, and Llama3.2-3B, respectively. We pick the 5 layers from each model for demonstration. In particular, we include the embedding layer, denoted as layer 0.

We can observe that all models exhibit the same trend as discussed in the main text, i.e., the Pearson's $r$ between $\Delta \log \pi(y_t|\mathbf{z})$ and $\nabla_\mathbf{z} \log \pi(y_t|\mathbf{z}_0)^\top \Delta\mathbf{z}$ decreases as the perturbation radius increases. The remainder term $\mathcal{O}(\|\mathbf{z}_1 - \mathbf{z}_0\|^2)$ increases accordingly with the perturbation radius increasing. Furthermore, as shown in Figure 5, 6, and 8, $\mathcal{O}(\|\mathbf{z}_1 - \mathbf{z}_0\|^2)$ increases more rapidly for smaller models than for larger ones, indicating that smaller models are less stable. Specifically, Llama3.2-1B shows a significant increase in variance as the perturbation radius increases. This suggests that Llama3.2-1B might be "unfamiliar" with portions of the data, indicating underfitting.

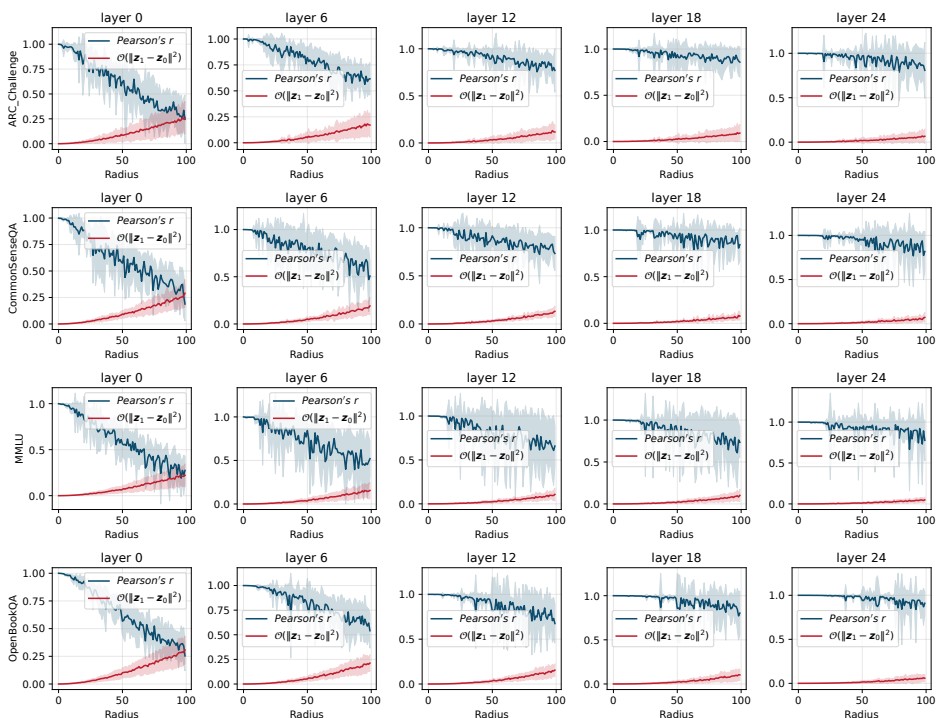

Figure 5: The Pearson's $r$ between $\Delta \log \pi(y_t|\mathbf{z})$ and $\nabla_\mathbf{z} \log \pi(y_t|\mathbf{z}_0)^\top \Delta\mathbf{z}$ of different layers of Qwen1.5-0.5B. The same trend is observed as in § 4.1.

### B.5 More Results about the Upper Bound

Next, we show more experimental results regarding upper bounds as mentioned in § 4.1. To compare the impact of different perturbation radius on the upper bound, we choose four perturbation radius for comparison: 0.1, 0.2, 0.4, and 0.8. Figure 10 shows the changes of the upper bound of $|\Delta \log \pi(y_t|\mathbf{z})|$ across LLMs' layers. We first observe that the gradient $\nabla_\mathbf{z} \log \pi(y_t|\mathbf{z}_0)$ exhibits a decreasing trend as the number of layers in the LLMs increases. The changes in upper bounds exhibit similar trends across LLMs within the same series, while showing slightly different trends between different series. For instance, the Qwen series shows gradients rising to a peak before steadily declining, only to surge dramatically at the end. In contrast, the Llama series exhibits gradients climbing to a peak followed by a continuous decline, with only a slight increase observed on the ARC Challenge and CommonSenseQA datasets. Large-scale models exhibit lower upper bounds than smaller ones, but unfortunately, the upper bounds of any model are insufficient to bring $\Delta \log \pi(y_t|\mathbf{z})$ close to 0.

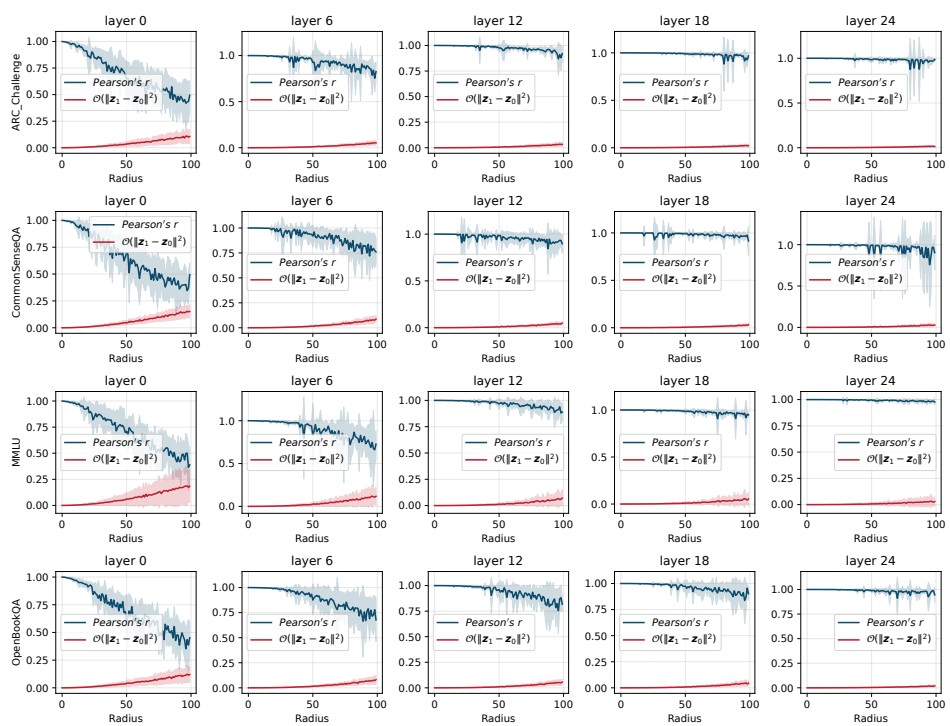

Figure 6: The Pearson's $r$ between $\Delta \log \pi(y_t|\mathbf{z})$ and $\nabla_{\mathbf{z}} \log \pi(y_t|\mathbf{z}_0)^\top \Delta \mathbf{z}$ of different layers of Qwen1.5-1.8B. The same trend is observed as in § 4.1.

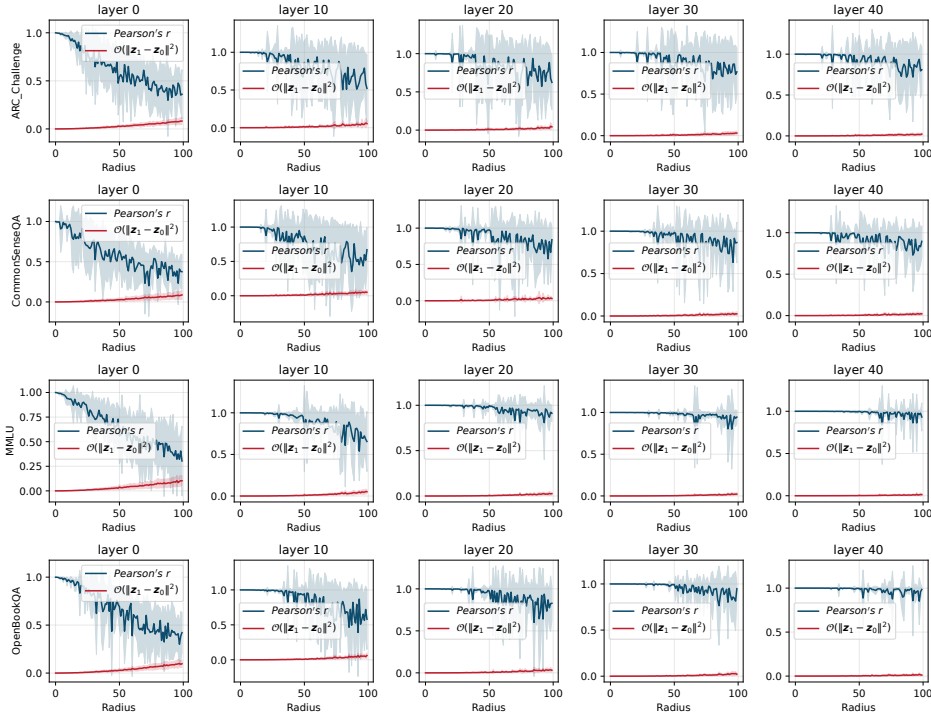

Figure 7: The Pearson's $r$ between $\Delta \log \pi(y_t|\mathbf{z})$ and $\nabla_{\mathbf{z}} \log \pi(y_t|\mathbf{z}_0)^\top \Delta \mathbf{z}$ of different layers of Qwen1.5-4B. The same trend is observed as in § 4.1.

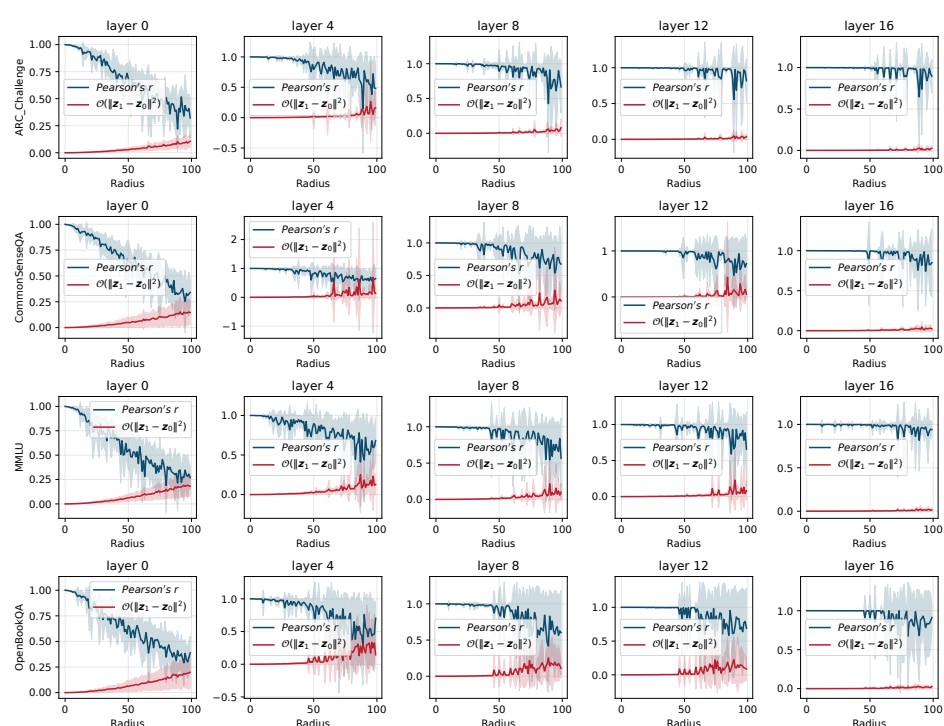

Figure 8: The Pearson's $r$ between $\Delta \log \pi(y_t|\mathbf{z})$ and $\nabla_{\mathbf{z}} \log \pi(y_t|\mathbf{z}_0)^{\top} \Delta\mathbf{z}$ of different layers of Llama3.2-1B. The same trend is observed as in § 4.1.

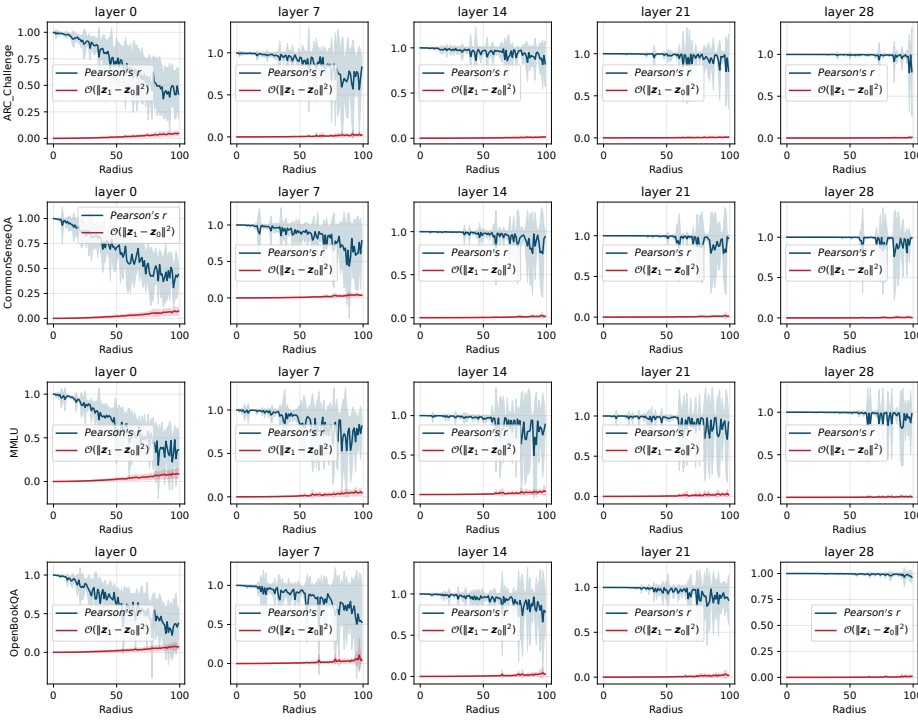

Figure 9: The Pearson's $r$ between $\Delta \log \pi(y_t|\mathbf{z})$ and $\nabla_{\mathbf{z}} \log \pi(y_t|\mathbf{z}_0)^{\top} \Delta\mathbf{z}$ of different layers of Llama3.2-3B. The same trend is observed as in § 4.1.

From Figure 10, we can observe that the gradient decreases as layers increase. Therefore, in Figure 11, we show the trends of another factor affecting the upper bound, $\|\Delta \mathbf{z}\|$. It is evident that $\|\Delta \mathbf{z}\|$ is always minimal at the embedding layer and then continues to increase. The differences in the increasing trends between the Qwen and Llama series are as follows: the Qwen series increases slowly at first, then grows sharply toward the latter half of the layers. The Llama series, however, increases sharply at the beginning, stabilizes, and then increases sharply again in the latter half of the layers. It is worth noting that relatively larger Qwen models (1.5-4B) attempt to decrease in the final layer, but the effect is far from sufficient. Overall, a larger perturbation radius leads to a larger upper bound.

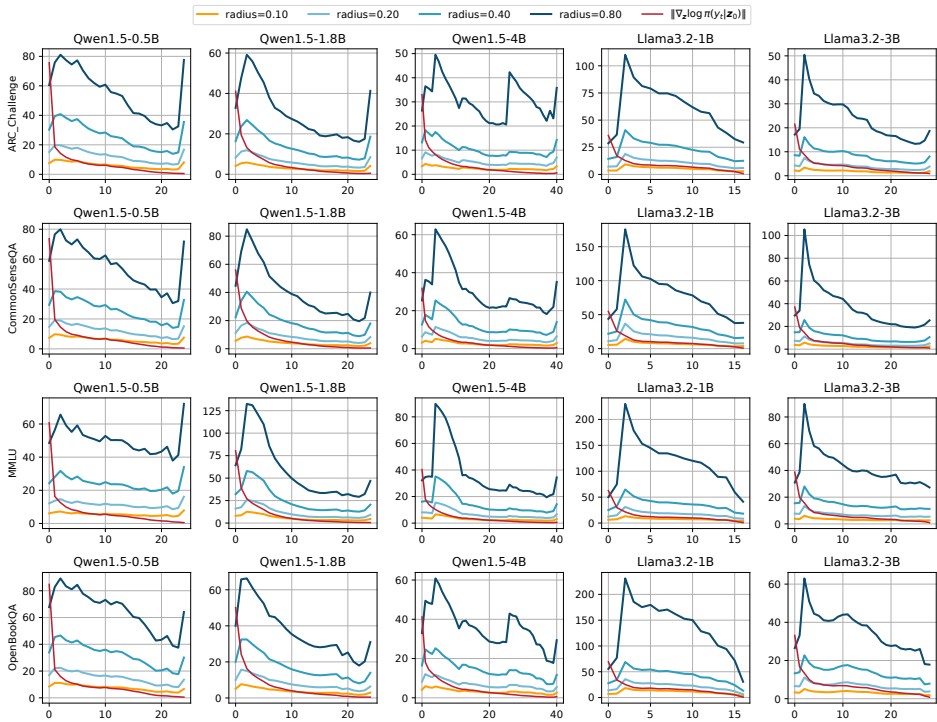

Figure 10: The changes of the upper bound of $|\Delta \log \pi(y_t|\mathbf{z})|$ across LLMs' layers.

## B.6 MORE RESULTS OF REAL-WORLD DATASET VALIDATION

In this section, we will continue from § 4.2 by first reporting additional experimental results for the *pad*, *trim*, and *equal* conditions (see § B.6.1). Subsequently, we will report further experimental results on how prompt modifications affect $\|\Delta \mathbf{z}\|$ (see § B.6.2). Finally, we will provide experimental details and more results for Figure 4d (see B.6.3).

### B.6.1 PAD VS. TRIM VS. EQUAL

We first perform experiments on all models across four datasets, calculating $\|\Delta \mathbf{z}\|$ for each layer and the number of tokens in the prompt. Subsequently, we group our experimental data based on *pad*, *trim*, and *equal* to calculate the average value for each layer. Figure 12 shows a comparison of *pad*, *trim*, and *equal* across all models and datasets. We can observe that $\|\Delta \mathbf{z}\|$ for *pad* and *trim* are generally higher than those for *equal* in most cases, indicating that our hard alignment method introduces some degree of bias. Moreover, counterintuitively, the gap between *pad*, *trim*, and *equal* is more significant on larger-scale models than on smaller ones. This serves as a cautionary note that larger models do not necessarily perform better, aligning with recent findings by Liu & Chu (2025). Specifically, the errors of Qwen1.5-0.5B and Qwen1.5-1.8B on the CommonSenseQA dataset are nearly negligible.

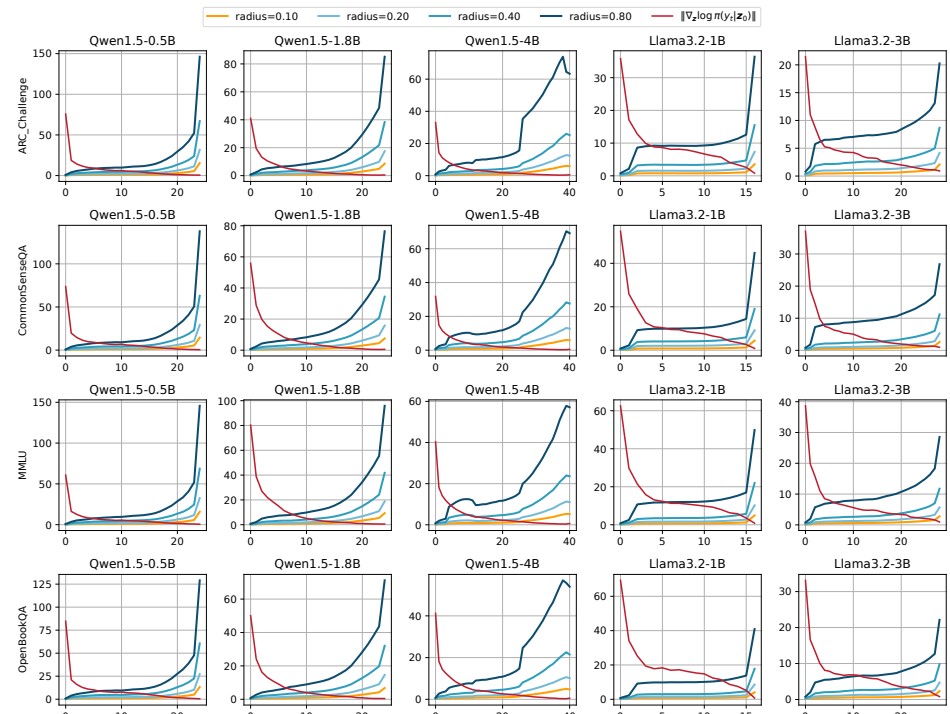

Figure 11: The changes of $\|\Delta\mathbf{z}\|$ across LLMs' layers.

### B.6.2 FIRST VS. LATTER & FEWER VS. MORE

In § 4.2, we already observed that modifying the first half of the prompt has a greater impact on $\|\Delta\mathbf{z}\|$ than modifying the latter half. More misalignments have a greater effect on $\|\Delta\mathbf{z}\|$ than fewer misalignments. In this section, we report experimental results across all models and datasets. By observing Figures 13 and 15, we conclude that the findings from § 4.2 hold true across all models and datasets, particularly becoming more pronounced in larger-scale models.

### B.6.3 PROMPT TEMPLATE VS. QUESTION

To investigate the relative impact of different prompt templates and questions on the output results (logits) of LLMs, we designed a two-factor ANOVA experiment. We selected questions from multiple datasets (ARC Challenge, CommonSenseQA, MMLU, OpenBookQA). We used the 12 prompts listed in Table 1. For each question, we input it with each of the 12 different prompt templates to obtain the model's logit outputs. To ensure a balanced design, we consistently selected the first 12 questions from each dataset. Combining these with the 12 prompts forms a 12×12 factorial combination, yielding a total of 144 experimental samples. We take question and prompt as classification factors respectively. Construct an ordinary least squares (OLS) regression model logit $\sim C(\text{question}) + C(\text{prompt})$. We perform an ANOVA on the model results to quantitatively evaluate the explanatory power of the question factor and prompt factor on logit variation. Simultaneously, we calculate the contribution ratio of each factor to the total sum of squares and compare it with the residual term.

As shown in Figure 15, we present the experimental results of all models across all datasets. It is not difficult to observe that, in all cases, prompts exert a greater influence on logits than questions do. In most cases, the prompt explains between 70% and 90% of the variance, while the question typically contributes less than 10%. The residual generally accounts for 5% to 20%, indicating that the primary variation in the model's output can be attributed to the prompt factor. This indicates that LLMs recognize prompt templates more than questions. This finding serves as a warning that when evaluating the performance of LLMs, the importance of prompt design may outweigh differences in dataset questions, posing challenges for fair model comparisons.

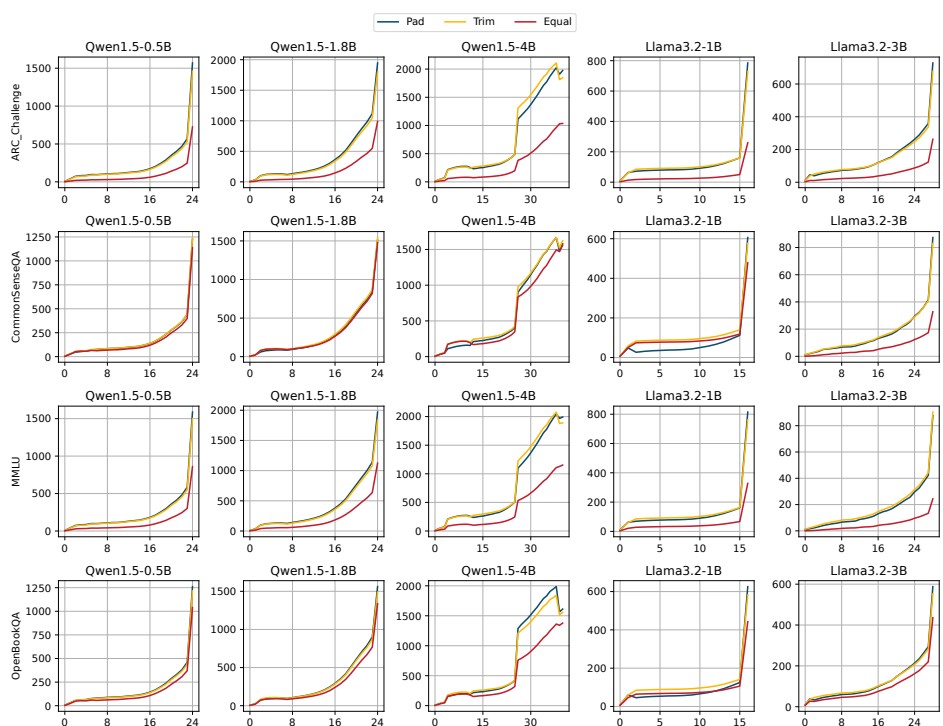

Figure 12: The biases caused by the hard alignment.

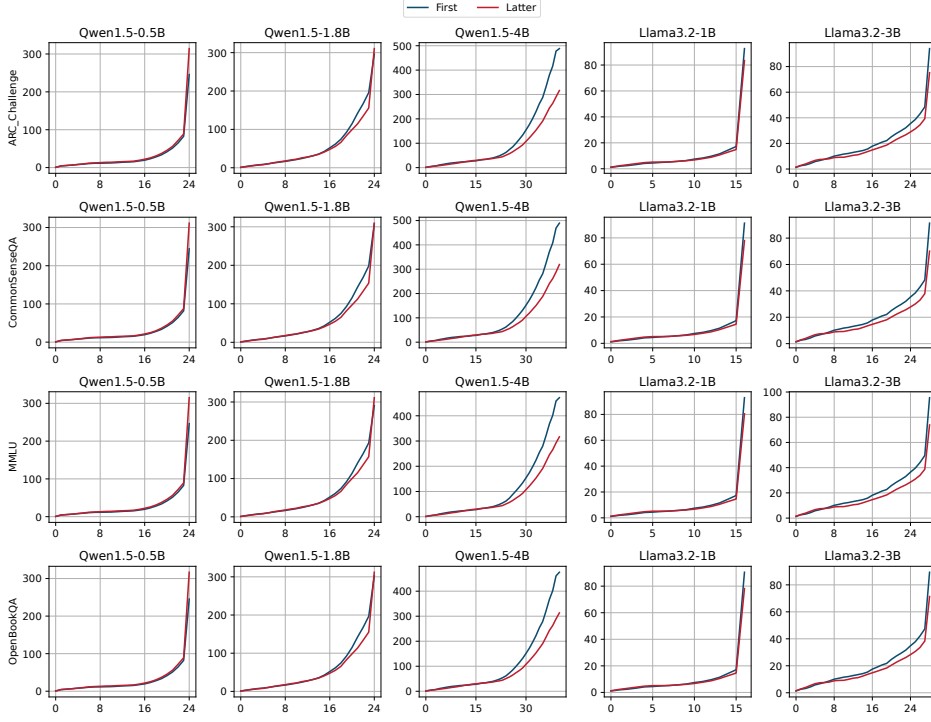

Figure 13: The comparison of $\|\Delta \mathbf{z}\|$ between modify the first half and latter half of the prompt.

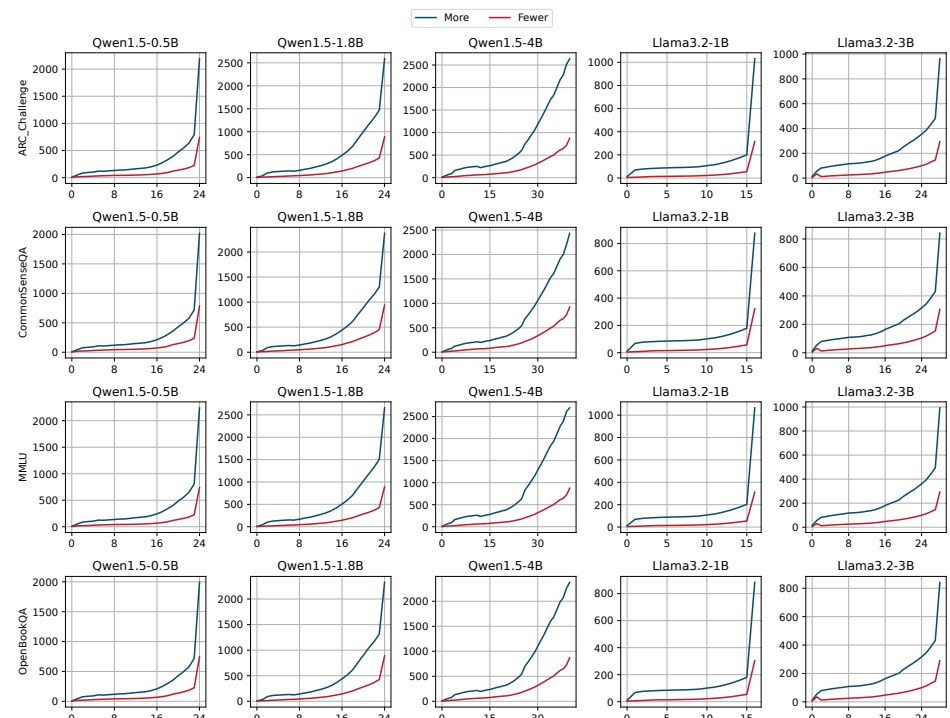

Figure 14: The comparison of $\|\Delta z\|$ between prompts with fewer and more misalignments.

## C  THE USE OF LLMS

To convey information more clearly, this paper employs `ChatGPT` to simplify certain complex expressions during the writing process.

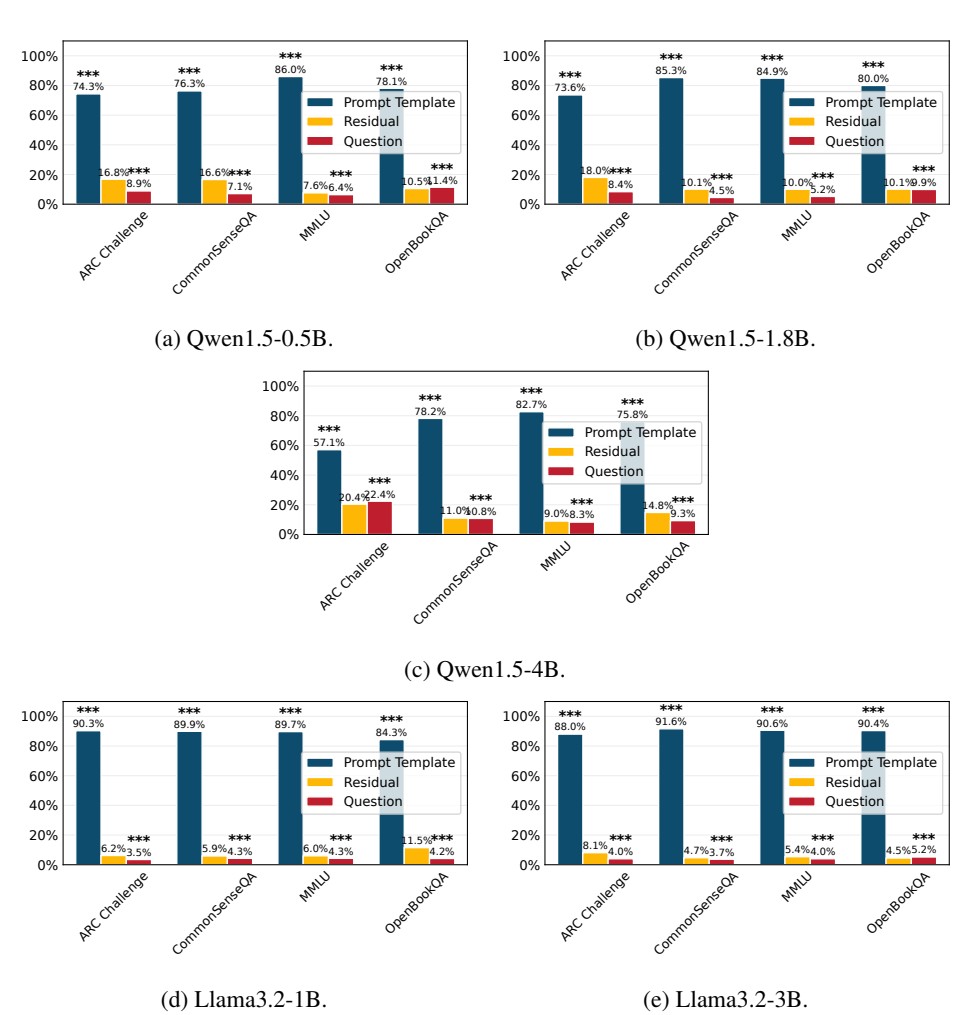

(a) Qwen1.5-0.5B.

(b) Qwen1.5-1.8B.

(c) Qwen1.5-4B.

(d) Llama3.2-1B.

(e) Llama3.2-3B.

Figure 15: The comparison of contributions from the prompt template and the question. The asterisks (***) indicate $p$-value$<0.001$.

