# OpenReview forum: "Understanding the Prompt Sensitivity"
_ICLR.cc/2026/Conference — ICLR 2026 Conference Withdrawn Submission_

### Official Review · Reviewer_Nwq3 · 2025-10-22

**Soundness:** 2
**Presentation:** 2
**Contribution:** 2
**Rating:** 2
**Confidence:** 4

**Summary:**

The paper investigates the sensitivity of large language models (LLMs) to user prompts through a Taylor series analysis of the model logits at two distinct prompt embeddings, z_0 and z_1. The analysis is motivated by an analogy to a ResNet trained on CIFAR-10, where features tend to cluster distinctly by class. The authors find that such clustering behavior is not consistently observed in LLMs.

The main issues with the paper include an overreliance on analogies from computer vision without sufficient consideration of the unique aspects of language modeling. In particular, a single meaning in natural language can correspond to multiple valid token sequences, which weakens the assumption that differences in logits at generated tokens reflect fundamental differences in LLM outputs. The methodology is also too simplistic; approximating the logits as linear via a first-order Taylor expansion is only valid in a very small neighborhood around the embedding z. As the distance between z_0 and z_1 increases, the linear approximation becomes unreliable, hence preventing it from capturing the more nuanced behavior of LLMs. Furthermore, the evaluation benchmarks are limited. The study considers only around 15 prompts and multiple-choice questions, which restricts the generality of the findings. Expanding the analysis to more diverse cases and broader sensitivity frameworks would make the results more representative.

**Strengths:**

- Prompt sensitivity is an interesting topic that can unlock new potential and improve prompt engineering strategies.
- The paper is well-written and generally easy to follow

**Weaknesses:**

- **Classification perspective**: The main premise of the analysis is motivated by an example of a ResNet trained on CIFAR-10. As shown by the authors in Figure 2, after training the ResNet, the final learned features become more clustered, with smaller distances between samples belonging to the same class. Building on this intuition, the authors aim to study whether similar behavior can be observed in LLMs. The problem is that the analysis focuses only on the generated token y_t, without considering the broader context of sequence generation. For instance, two prompts may produce different first tokens yet converge to nearly identical sequences afterward (e.g., one prompt causes the model to first output a newline before the answer, while another goes straight to the answer). Sensitivity metrics such as POSIX [Ref] account for this by evaluating the similarity across the entire generated sequence rather than focusing solely on \pi(y_t|**z**). This omission limits the validity of the authors’ analysis, as it overlooks the sequential dependencies intrinsic to language generation.

An example also can be seen in the “You are a helpful assistant…” case (Prompt 7 in Appendix B.2). The model might tend to output “Yes, of course” to appear helpful before providing the actual answer. Meanwhile, Prompt 9, which lacks the “Answer:” suffix, may cause the model to output that word itself. For these reasons, comparing full output sequences rather than single tokens would provide a more accurate and meaningful measure of sensitivity.

- **Contributions**: The paper’s contribution centers solely on analyzing prompt sensitivity, without proposing any method to address or mitigate it. This places high expectations on the depth and rigor of the analysis, which remains simplistic. The analysis relies on a Taylor expansion and a linear approximation of the logits around a neighborhood of z. Such an approach is only valid within very small neighborhoods (Equation 2 holds only as z_1 approaches z_0). Consequently, it fails to capture the broader nonlinear behavior of LLMs. For instance, prompts that are embedded farther apart in the latent space may still produce similar outputs, thus violating the assumptions of the linear approximation.

- **Presentation**: A large portion of the paper is dedicated to basic explanations of neural networks and transformers, which makes it read more like a tutorial. The space could be used more effectively by condensing the introductory material (e.g., Section 3.1, which reintroduces how transformers work) and focusing more on the depth of the analysis and put these details in the appendices.

- Minor comment: It would be beneficial to extend the sensitivity analysis to include variations caused by typos, punctuation, and related linguistic noise, similar to what was performed in [Ref].

[Ref]: POSIX: A Prompt Sensitivity Index for Large Language Models (Chatterjee et al., Findings of EMNLP 2024) — https://aclanthology.org/2024.findings-emnlp.852/

**Questions:**

Please refer to the weaknesses section.

---

### Official Review · Reviewer_fjhi · 2025-10-27

**Soundness:** 2
**Presentation:** 3
**Contribution:** 2
**Rating:** 4
**Confidence:** 4

**Summary:**

This paper investigates prompt sensitivity in large language models using a first-order Taylor expansion framework. It concludes that this sensitivity stems from the lack of internal clustering of semantically similar inputs, leading to dispersed hidden states and consequently a large upper bound for output logit differences that rarely approaches zero. These findings are supported through both perturbation analysis and real-world dataset validation. While the conclusions are interesting, the technical contribution and practical applicability of the proposed approach remain unclear.

**Strengths:**

S1) The paper's finding that the core reason for prompt sensitivity lies not in high gradients but in how LLMs inherently disperse the hidden states of semantically similar prompts thereby preventing their outputs from converging is interesting.
S2) The paper is easy to follow due to its clear structure and straightforward explanations. The concepts are presented in plain language with consistent terminology throughout.
S3) The authors have put significant effort into ensuring reproducibility by providing code and detailed experimental configurations.

**Weaknesses:**

W1) The core analytical tool in this work is a first-order Taylor expansion used to approximate differences in model outputs. It represents a generic technique applicable to any differentiable model rather than providing insights specific to LLM architectures. I do not see a clear technical distinction between this approach and existing methods such as [1]. The upper bound derived by the authors does not adequately explain why LLMs exhibit greater prompt sensitivity compared to traditional neural network architectures. The conclusion that sensitivity stems from a lack of internal clustering of semantically similar inputs appears to be drawn primarily from experimental results, rather than emerging naturally from the proposed theoretical framework. As such, I find the contribution of the interpretation method itself to be limited.

W2) The proposed method closely resembles a highly simplified version of Integrated Gradients (IG) [2]. When considering the embedding as the function's input, the approach reduces to a degraded form of IG. Whereas IG properly captures model behavior by integrating gradients along the entire path from a baseline input, this work relies solely on the gradient at a single starting point, performing a crude linear extrapolation to estimate behavior at the endpoint. I think the author should discuss the advantage of the proposed method over existing gradient-based interpretation methods.

W3) Conventional interpretability methods aim to attribute model behavior to specific input features, providing users with intuitive insights they can act upon. For example, if a loan application is rejected due to employment status, the applicant can address this by finding a job to improve future approval chances. In contrast, while this paper identifies LLM prompt sensitivity as the problem, it's unclear how the proposed method actually helps mitigate it. The analysis may reveal that sensitivity arises from dispersion in hidden states, but what practical steps does it enable? How does this theoretical insight translate into concrete improvements in prompt design or model behavior? The connection between diagnosis and solution remains unestablished.

[1] Montavon G, Lapuschkin S, Binder A, et al. Explaining nonlinear classification decisions with deep taylor decomposition[J]. Pattern recognition, 2017, 65: 211-222.
[2] Sundararajan M, Taly A, Yan Q. Axiomatic attribution for deep networks[C]//International conference on machine learning. PMLR, 2017: 3319-3328.

**Questions:**

N/A

---

### Official Review · Reviewer_3saz · 2025-10-28

**Soundness:** 2
**Presentation:** 3
**Contribution:** 1
**Rating:** 2
**Confidence:** 3

**Summary:**

This work investigates the prompt sensitivity of large language models, i.e. why they produce different outputs for prompts that have the same meaning. The authors treat LLMs as multivariable functions and use a first-order Taylor expansion to analyze how small changes in prompt embeddings affect output logits. They find that, unlike traditional neural network architectures such as ResNets, LLMs do not cluster semantically similar inputs in their internal representations. They interpret this as a potential reason why close input prompts can give very different outputs.

**Strengths:**

Prompt sensitivity is a significant and timely issue, and up to my knowledge, we are still lacking a good theoretical framework to study it. Moreover, the paper is well written and easy to follow.

**Weaknesses:**

Although the topic is highly interesting, I found that this work lacks sufficient novelty and depth in its current form. Employing first-order methods to analyse model input sensitivity is a well-established approach. Besides, as the authors acknowledge, the absence of clustering for semantically similar inputs in LLMs compromises the applicability of the first-order approximation used in their analysis.

A few typos:
- Eq 6: summation should likely be over $x_i \not = x_j$
- lines 201-202: I think it should be L x D instead of L x N
- Eq. 10: the statement is somewhat imprecise, the upper bound only holds within a neighbourhood of $z_0$, and even then, counter-examples can be constructed where it fails for certain $z_1$. For instance, if the Hessian is positive definite, the inequality would not hold when $z_1 - z_0$ aligns with the gradient direction.

**Questions:**

Given that the paper shows LLMs do not cluster semantically similar inputs, what would be a meaningful analytical tool to analyse prompt sensitivity ?

---

### Official Review · Reviewer_TwcC · 2025-10-30

**Soundness:** 2
**Presentation:** 3
**Contribution:** 2
**Rating:** 4
**Confidence:** 4

**Summary:**

This paper tries to explain a key problem: why are LLMs so sensitive to their prompts? The authors use Taylor expansion to argue that this sensitivity comes from the model's failure to "cluster" similar inputs internally, a behavior they call "dispersion."

While the paper tackles a very important topic and has some well-run experiments, I am confised to its central argument. Its main analytical tool isn't proven to work in real world, which makes the paper's core explanation not very convincing.

**Strengths:**

+ It's a fresh idea to use Taylor expansion to analyze LLM's local behavior. This approach provides a formal language (gradients, hidden state distances) to talk about the abstract problem of "sensitivity," and this effort to bring mathematical structure to the problem is valuable.

+ The experimental work here is impressive. For example, in the perturbation analysis, the way the authors systematically vary the radius and track the Pearson's r is a very solid and well-designed method to test the limits of their theory. Also, when looking at real-world prompts, their use of Analysis of Variance (ANOVA) with strong statistical significance (p<0.001) adds real weight to their empirical findings. This process shows a commitment to rigorous validation.

**Weaknesses:**

**1. One Argument is Built on a Flawed Comparison.**

One of the issue is with the paper's starting point. It builds its argument on the idea that LLMs should behave like a ResNet image classifier, which learns to group similar things together. A classifier's job is to achieve **invariance**. An LLM's job is often the exact opposite: it *must* pay close attention to tiny changes in word order and phrasing. So, the "dispersion" the paper finds might not be a bug, but a **feature**—a sign the model is correctly doing its job of being sensitive to sequence structure.

**2. A Gap Between the Theory and Reality.**

The paper's main analytical tool (the Taylor expansion) only works well when the changes to the model's hidden states are tiny. The authors do a good job of showing this works for their own small, artificial perturbations.

However, their own results show that real-world prompt templates create changes (`||Δz||`) that are *much larger* than these artificial ones. The paper didn't show shows us if its Taylor expansion is still a good approximation in this real-world scenario. It's missing the key experiment: what is the Pearson's r for actual prompt pairs? Without that proof, we can't trust that the theory actually explains the real problem it set out to solve.

**3. The Explanation for Sensitivity Feels a Bit Obvious.**

The paper's main takeaway is: LLMs are sensitive on the outside because their internal representations are also sensitive to changes. I feel it is a bit obvious. We would expect any deep, non-linear network to behave this way if it is sensitive: a different input leads to more different internal states, which in turn leads to a more different output. While the paper quantifies this, the "why" part of the explanation feels circular.

**4. The Analysis Stops Short of the Final Output.**

A smaller, but still important, point is that the analysis stops at logits. The final word we actually see is chosen after a softmax and argmax step. A huge change in logits might not change the winning word if one option is already far ahead, while a tiny change could flip the result between two close contenders. By not discussing this last crucial step, the analysis feels incomplete.

**Questions:**

Please refer to weaknesses.

---

### Note · Authors · 2025-12-26

I have read and agree with the venue's withdrawal policy on behalf of myself and my co-authors.